# Precision Medicine Care in ADHD: The Case for Neural Excitation and Inhibition

**DOI:** 10.3390/brainsci11010091

**Published:** 2021-01-13

**Authors:** Ping C. Mamiya, Anne B. Arnett, Mark A. Stein

**Affiliations:** 1Institute for Learning and Brain Sciences, University of Washington, Seattle, WA 98195, USA; 2Department of Psychiatry & Behavioral Sciences, University of Washington, Seattle, WA 98195, USA; arnettab@uw.edu (A.B.A.); mark.stein@seattlechildrens.org (M.A.S.)

**Keywords:** amphetamine, inhibitory control, methylphenidate, executive function, dopamine, norepinephrine

## Abstract

Attention-deficit/hyperactivity disorder (ADHD) is a neurodevelopmental disorder that has become increasingly prevalent worldwide. Its core symptoms, including difficulties regulating attention, activity level, and impulses, appear in early childhood and can persist throughout the lifespan. Current pharmacological options targeting catecholamine neurotransmissions have effectively alleviated symptoms in some, but not all affected individuals, leaving clinicians to implement trial-and-error approach to treatment. In this review, we discuss recent experimental evidence from both preclinical and human studies that suggest imbalance of excitation/inhibition (E/I) in the fronto-striatal circuitry during early development may lead to enduring neuroanatomical abnormality of the circuitry, causing persistence of ADHD symptoms in adulthood. We propose a model of precision medicine care that includes E/I balance as a candidate biomarker for ADHD, development of GABA-modulating medications, and use of magnetic resonance spectroscopy and scalp electrophysiology methods to monitor the effects of treatments on shifting E/I balance throughout the lifespan.

## 1. Introduction: Need for Precision Medicine Care in ADHD

Attention-deficit/hyperactivity disorder (ADHD) is a common, impairing neurodevelopmental disorder affecting more than six million children in the United States (https://www.cdc.gov/ncbddd/adhd/data.html), many of whom will continue to experience symptoms and associated dysfunction well into adolescence and adulthood [1]. In addition to core symptoms of ADHD, individuals are at increased risk of psychiatric comorbidity, including substance use, and impaired academic, occupational, and health maintenance [2,3]. Moreover, societal costs in the United States of the disorder are estimated to exceed 124 billion dollars [4]. 

Patients with ADHD and their clinical providers are challenged with the considerable heterogeneity in clinical presentation, symptom trajectory, and treatment response [5]. Currently, there are multiple pharmacological treatment options, including derivatives of methylphenidate and amphetamine stimulant classes, which target dopamine transporters, as well as several non-stimulants, which act as receptor agonists for norepinephrine and epinephrine systems [6]. However, critical limitations to ADHD treatment remain: approximately 25% of children with ADHD are stimulant “non-responders” and many individuals experience intolerable side effects of these medications and discontinue treatment despite persistent symptoms [7,8]. Clinical subtypes of ADHD (e.g., predominantly inattentive, versus predominantly hyperactive/impulsive, versus combined presentations) are not predictive of treatment response [9,10], likely due to poor test–retest reliability of these categorizations [11]. Known etiological subtypes of ADHD, such as prenatal exposure to alcohol, may be associated with differential response to stimulant classes, but there nonetheless remains substantial heterogeneity in treatment response within these populations [12]. Clinicians are thus left to employ a trial-and-error approach to treatment planning, which can lead patients to experience medication trial fatigue before identifying their most effective agent and dose. Alternatively, a precision medicine care could be developed and evaluated where biomarkers and clinical features could guide treatment planning at the individual level. 

There is a vast collection of extant literature on neurobiological correlates of ADHD that could ultimately inform a precision medicine approach [13,14]. In this review, we focus on deficient fronto-striatal connectivity driven by atypical balance of neural excitation and inhibition (E/I). We propose that E/I imbalance impacts the development of neural circuitry and organization of synaptic connections in developing brains. Over the course of development, these effects could lead to atypical brain structural and functional measures in adolescents and adults with ADHD. We review literature that strongly implicates GABA and glutamate as contributing to E/I imbalance among individuals with ADHD. Finally, we consider methods for precise measurement of E/I balance in the prefrontal cortex (PFC) and striatum, as a biomarker that could inform precision medicine care. Altogether, we aim to review evidence that E/I imbalance contributes to atypical neurodevelopment, and we consider support for this pathogenesis as a potential treatment target for children and adolescents with ADHD (Figure 1).

## 2. Imbalance of E/I in ADHD

Neurotransmitter Systems: Decades of literature point to catecholamine (i.e., dopamine and norepinephrine) deficiency among individuals with ADHD [15,16,17]. ADHD symptoms are associated with response to dopamine agonists, intracellular dopamine receptor availability, and dopamine transporter genotypes [18,19,20]. Moreover, dopamine levels in the PFC-striatal circuitry have been linked to behaviors that are not core clinical symptoms of ADHD, but are nonetheless associated with the disorder, such as risk-taking [21,22] and delay discounting [23,24]. As such, over 60% of children and adolescents diagnosed with ADHD are treated with stimulant medications [25]. First line pharmacological treatments for ADHD include methylphenidate and amphetamine. The mechanism of methylphenidate and amphetamine treatments is to block dopamine and norepinephrine transporters, thus preventing removal of both molecules from the synapse. In addition, amphetamine has been shown to reverse transporter activity, causing increased dopamine efflux. In the human brain, striatum has the highest dopamine transporter concentrations. Thus, the principal effect of methylphenidate and amphetamine is to increase synaptic dopamine and norepinephrine concentrations in the striatum [26].

There is emerging evidence that GABA and glutamate levels and their ratio contribute to the neurobiological etiology of ADHD and related neurodevelopmental disorders [27,28]. Several factors including GABA, orthodenticle homeobox (Otx2), neuronal pentraxins (NARP), neurotrophins (BDNF), and neuregulin impact developing brains through a variety of regulatory mechanisms, from myelination to synaptic communications [29,30,31,32,33,34]. Computational studies have shown that these intrinsic factors program the maturation of interneurons [30], thus allowing strong GABA-mediated feedforward inhibition to maintain the stability of local neural networks [35]. 

Resting-state GABA and glutamate concentrations in the frontal, striatal and premotor areas have been consistently reported as altered among children and adults with ADHD. Children with ADHD show reduced GABA concentrations in the motor cortices [36], striatum [27], and the thalamus. In contrast, glutamate concentrations appear to be elevated in the prefrontal and anterior cingulate cortices [37,38], and decreases with age [39]. In adults with ADHD, both GABA and glutamate concentrations are reduced in the anterior cingulate cortex and the striatum [40,41,42], but also, see [43]. Altogether, these studies suggest that GABA and glutamate concentrations in the fronto-striatal circuitry undergo different developmental trajectories. 

Importantly, GABA and glutamate concentrations are heavily influenced by the time of day, arousal state, and environmental changes [44,45,46,47]. The homeostasis of GABA and glutamate concentrations is tightly calibrated in the biosynthetic process and is highly dependent on neural activity [48,49,50]. Once released from neurons, GABA and glutamate transporters are the primary source of clearing these two key brain metabolites from the synaptic clefts and uptaking them to astrocytes [51,52,53]. Thus, neural circuitry involved in cognitive control of attention will most likely exhibit labile E/I balance when people are directing their attention. We recently demonstrated that adults with ADHD showed reduced GABA and glutamate concentrations in the ACC and the caudate nucleus while people directed their attention to monitor and report conflicts [54]. Altogether, this body of research indicates that the balance of GABA and glutamate signaling in the fronto-striatal circuitry is crucial for attention control, and deficient GABA signaling in this circuitry may be related to impaired attention control in ADHD. Moreover, these reports suggest that glutamate–glutamine (GABA) cycling is likely disrupted throughout the lifespan in this population.

Spectral Slope: Not surprisingly, alterations in GABA inhibition are strongly implicated in abnormal neural oscillations marked by E/I imbalance. In neurophysiology studies, neural oscillation at gamma frequency range is highly correlated with GABA content in healthy and diseased individuals [55,56]. The inverse association between power and frequency of neural oscillations in the brain can be summarized as a 1/f distribution (also known as spectral slope), whose scale-free property is one of the key principles of neural organization [57]. The spectral slope varies in response to shifting cognitive demands and cortical arousal [58]. More importantly, the spectral (1/f) slope has been linked to E/I balance in animal studies: Gao and colleagues demonstrated that reduced E/I ratio in macaque and rat cortices is correlated with steeper spectral slope, and that this can be accounted for by greater GABA_A_ synapse density [59]. A large body of research has reported that increased theta/beta ratio is a neural signature of ADHD [60,61], which would be consistent with a steeper spectral slope. However, variability in the reported effect sizes for this ratio has led researchers to propose that this finding may be explained by steeper spectral slope in the ADHD group, corresponding to reduced E/I ratio [62]. Only two studies have directly investigated the association between spectral slope and ADHD. Robertson and colleagues [62] found a steeper slope among children with ADHD at rest, while Pertermann and colleagues found no difference between unmedicated ADHD and control children during a cognitive task [63]. Thus, spectral slope as a candidate biomarker for ADHD remains uncertain and deserves further investigation.

Limitations: To date, there is no single genetic variant [64], neurocognitive profile [65], or other reliable biomarker for ADHD. As candidates for ADHD biomarker measurement, MR spectroscopy and EEG present with strengths and limitations. Owing to recent improvements in GABA-editing pulse and higher field strength MR scanners (>1.5 T), the detection of GABA and glutamate concentrations in the cortex and subcortex can be achieved with high precision in the developing and mature brains [66,67]. These technical advancements open up the potential of using GABA and glutamate concentrations as biomarkers during the pathogenetic stages of ADHD. However, MRS-based quantifications are the summation of these molecules from various compartments within a brain imaging voxel, including intracellular and extracellular space. Thus, they provide a gross quantification of glutamate and GABA concentrations and are critically determined by signal-to-noise ratio and susceptible to any movements during the scan. The interpretations of MRS-based glutamate/GABA ratio and its relation with neural activity at the synaptic level requires further investigations. 

EEG is a non-invasive, relatively low-cost method for measuring neural activity and therefore presents as a highly attractive method for measuring E/I in clinical patients. Moreover, many research EEG systems involve easy application of gel-free electrode caps, making it a feasible tool for use in outpatient settings. On the other hand, as with other neuroimaging methodologies, scalp electrophysiology is sensitive to artifact sources. In particular, muscle movement, cardiac signal, and power lines artifacts require data cleaning steps that may distort the original signal and introduce a degree of error. The use of multiple, repeated trials reduces part of this error, but continued biomarker development would benefit from confirmation with other methodologies [68]. For instance, given the low spatial resolution of EEG, methods that access closed-field electrogenic sources, such as MRI, would complement the methods described here by providing a structural perspective of the biological mechanisms. Whereas EEG spectral slope reflects an additive signal derived from extracellular fields with multiple neuronal sources, the strength of which is attenuated as it is dispersed across the scalp [57], MRI can confirm at the level of neuronal circuits. Many of these limitations could be addressed through a longitudinal approach to biomarker measurement, where intra-individual change over time, for example, in spectral slope, constitutes the biomarker, rather than one measurement at a single moment in time. 

## 3. Developmental Sequelae of E/I Imbalance: Altered Brain Connectivity 

Neural E/I balance is critical to synaptic plasticity and neuronal growth and pruning. The maturation of the nervous system is shaped by activity-dependent tuning of synaptic connections and large-scale regional connections between neural networks. Within local networks in particular, GABA-releasing synapses sculpt neural connections in developing brains [69] and stabilize neural activation in mature brains [30]. Thus, childhood constitutes a critical developmental period during which GABA signals sharpen the signal-to-noise ratio in local neural circuits, and enable effective signal transductions through long-range neural connections [70,71]. We propose that altered E/I balance in early development of children with ADHD leads to atypical brain connectivity, which explains long-term persistence of symptoms among the majority of patients. In the following section, we describe developmental research that supports E/I imbalance in childhood and subsequent atypical neurodevelopment.

The administration of stimulant medication is shown to normalize atypical brain connectivity measures among children with ADHD [14,72,73,74]. In particular, children who show good response to stimulant treatment exhibit elevated theta/beta ratio compared to poor responders [75]. Similarly, the effect of stimulant treatment normalizes the absolute theta and delta power in adults with ADHD [76]. MR imaging analyses provide anatomical and neurochemical evidence that details the effects of stimulant treatment on brain connectivity in the fronto-striatal circuitry. Using functional MR imaging, children receive long-term treatment of methylphenidate are shown to exhibit normalized brain connectivity resembling that of neutrally typical individual children [77,78,79,80]. Moreover, recent evidence shows that stimulant medication increases GABA concentrations in the medial prefrontal cortex only in adults with ADHD who receive stimulant treatment before, but not after adulthood [27]. This apparent age-related stimulant treatment effect on brain connectivity suggests that childhood may constitute a sensitive period during which optimization of neuronal E/I balance could facilitate healthy neural growth and organization. Accordingly, longer duration of stimulant treatment has been associated with greater normalization of fronto-striatal white matter connectivity among adolescents with ADHD [81]. Moreover, a meta-analysis of structural imaging studies of adults with ADHD found that prior stimulant treatment was associated with reduced differences in anterior cingulate volume [82]. Altogether, this body of research supports the hypothesis that pharmacological treatment of ADHD in childhood may facilitate healthy brain growth by correcting imbalance of neural E/I. 

Neuroimaging research is consistent with altered connectivity of brain networks in ADHD, particularly in the PFC and striatum [83]. Recent evidence from ENIGMA-ADHD collaboration and other large case-control studies provide evidence that connectivity in the fronto-striatal circuitry is reduced in adolescents and adults, but not children, with ADHD [71,84,85]. This stark contrast between childhood and adulthood ADHD may be explained by compensatory mechanisms employed throughout development. Indeed, Saad, et al. [86] reported that children and adolescents with ADHD show more neurite pathways extending from the mid-PFC, hippocampus, anterior cingulate cortex (ACC), amygdala, and putamen compared to neural typical individuals, suggesting decreased axonal pruning in early development. Importantly, these brain regions largely overlap with areas in which reduced cortical volume has been found among children and adults with ADHD [87,88], supporting the notion that increased number of neurite pathways may be compensatory for reduced brain volume. EEG coherence, which is a measure of oscillatory synchrony across disparate electrodes, is generally elevated among individuals with ADHD, likewise indicating greater connectivity in this group, particularly in frontal-parietal scalp regions [89,90]. 

Longitudinal studies indicate atypical long-term neural connectivity among individuals with ADHD. In a large, prospective study of youth, Kessler and colleagues [91] found that ADHD diagnosis was associated with less steep linear growth in neural connectivity, rather than a maturational delay that would eventually normalize. Mattfield and colleagues [92] reported that persistence of ADHD symptoms in adulthood was associated with reduced default mode network (DMN) connectivity, as well as reduced de-coupling of the DMN from functional brain networks during active tasks. In contrast, an EEG study found no differences in EEG coherence (i.e., oscillatory synchrony across disparate electrodes) between young adults with persistent versus remittent ADHD symptoms [93]. Rather, both groups showed a pattern of hyperconnectivity compared to controls, consistent with the broader literature [89,90]. Thus, ADHD appears to be associated with long-term consequences for neural network organization, with greater neural connective dysfunction indicating higher likelihood of symptom persistence across the lifespan.

## 4. GABA and Glutamate Modulating Treatments in ADHD and Related Neurodevelopmental Disorders

Catecholamine agonists are believed to increase the E/I ratio by increasing availability of excitatory neurotransmitters [94]. However, as noted above, the success of these agents in treating ADHD is counterbalanced by the proportion of patients who show poor response or intolerable side effects to stimulant medications. Therefore, development of medications that target alternative neurotransmitter systems, such as GABA and glutamate, has been proposed [95]. 

GABA- and glutamate-modulators, such as arbaclofen, bumetanide, and tiagabine, have already been tested for treatment of neurodevelopmental disorders that frequently co-occur with ADHD, such as epilepsy [96], autism spectrum disorder (ASD) [97], and Fragile X syndrome [98]. Patterns of comorbidity and common genetic factors among these disorders suggest shared neurobiological mechanisms associated with neural E/I imbalance [99]. Moreover, individuals with ADHD have high rates of autism symptoms, such as social difficulties, sensory seeking, and restricted interests [100]. Both GABA receptor B agonists like arbaclofen, and GABA reuptake inhibitor, such as tiagabine, have long been used to treat epilepsy with a high level of success [101]. Their anti-epileptic effect is to augment inhibitory neurotransmission and increase GABA availability in the synapse. Animal studies have indicated that GABA agonists can improve ASD-associated behaviors in rat models of ASD and Fragile-X syndrome [102,103]. On the other hand, human clinical trials have demonstrated mixed results [104]. For example, Lemonnier and colleagues [105] reported that bumetanide, a potent sodium-potassium-chloride cotransporter inhibitor, improves social communication and restrictive/repetitive behaviors among children and adolescents with ASD after three months of treatment. Another recent study found that behavioral improvement associated with bumetanide was associated with restored GABA/glutamate ratio in the insular and visual cortices of children with ASD [106]. However, clinical trials of arbaclofen have failed to find reduction of core ASD symptoms among youth with ASD and/or Fragile X syndrome, despite improvements in secondary endpoints, such as irritability and adaptive functioning [107,108]. 

At the time of this writing, there are few published trials of GABAergic and glutamatergic treatments targeting ADHD symptoms. Long-term treatment of an oral GABA supplement, Gamalate@B6, has been reported to decrease hyperactivity and improve attention in adults with borderline to mild intellectual disability and coexisting ADHD [109]. In another study, Org 26576 (8,9,9a,10-(S)-tetrahydro-5H, 7H-Pyrido-[3,2-f]pyrrolo [2,1-c][1,4] oxazepin-5-one) significantly reduced locomotor activity. Org 26576 modulates the ionotropic AMPA-type glutamate receptor and enhances excitatory neurotransmission in the nervous system. A fixed dose of Org also effectively reduced clinical symptoms by increasing AMPA-receptor mediated signaling, and potentially, by increasing dopamine and norepinephrine in the medial prefrontal cortex [110]. Gene network analysis consisting of eight metabotropic glutamate receptor (mGluR) subtypes reveals that adolescents with ADHD and a disruptive genetic variation within the network showed reduced severity of ADHD symptoms, as well as improved overall functioning, following five weeks of administration of a non-selective mGluR activator, NFC-1 [111]. Finally, one study demonstrated that atomoxetine, a norepinephrine reuptake inhibitor, increased neuronal excitation in the PFC of rats [112], suggesting that this non-stimulant treatment may not only increase norepinephrine levels, but may also affect glutamate signaling in the prefrontal cortex.

While research on direct effects of GABAergic modulators on ADHD symptoms is limited, there is ample evidence to suggest that catecholamine-targeting stimulant medications may simultaneously affect GABAergic neurotransmission. For example, mouse brain slice research has shown that extracellular GABA in the striatum can be brought into dopaminergic axonal terminals by vesicular monoamine transporters [113,114]; when midbrain dopaminergic neurons are excited, striatal dopamine terminals co-release dopamine and GABA, subsequently exciting GABA receptors in the same region. Accordingly, common stimulant medications for ADHD have been shown to increase GABA neurotransmitter levels and receptor activation in the PFC-striatal circuitry [115,116]. Thus, as alternative therapies for ADHD are developed, it is critical to acknowledge that the neural circuits and release mechanisms associated with GABA/glutamate and catecholamines are not entirely distinct. Rather, we propose that the effects of these neurotransmitter systems on healthy E/I balance in the developing brain are both additive and interactive. 

## 5. Conclusions and Future Directions: Implications for Precision Medicine Care

The literature reviewed here supports a model in which interactions among GABA, glutamate, and catecholamine neurotransmissions underlie E/I imbalance in the brains of very young children with ADHD. This imbalance likely propagates atypical neural development and network connectivity, leading to long-term ADHD symptom persistence and associated psychopathology. Structural and functional brain abnormalities reported among adolescents and adults with ADHD may be understood as a consequence of reduced synaptic signaling during experience-dependent stages of early neurodevelopment. However, due to the lack of prospective studies of E/I balance in infants who go on to develop ADHD, we cannot fully rule out that ADHD-like behaviors lead to E/I imbalance in later development. We find hope in reports that indicate this developmental cascade can be at least partially normalized by correction of E/I imbalance via administration of psychostimulant medications, even into adulthood. However, individual responses to stimulant and non-stimulant interventions vary considerably; thus, it remains a challenge for medical caregivers to predict treatment effectiveness without reliable biomarkers or clear demographic indicators [7]. 

Precision medicine care depends upon two critical components: (1) availability of multiple treatment options; and (2) identification of reliable, valid biomarkers that map onto these available interventions. In ADHD, the first goal has at least partially been achieved. Multiple psychostimulant classes and non-stimulants exist, each offering slight variations in neurotransmitter targets, mechanisms, and duration of effect. However, side effects and tolerability restrict the impact and effectiveness of current treatments with few predictors of adverse events. The development of GABAergic and glutamatergic medications show promise for ADHD as well as other neurodevelopmental disorders, and merit continued investigation, particularly among children who do not tolerate stimulant medications well or non-responders. 

With respect to the second criteria for precision medicine, to date, traditional neuroimaging and genetic methods have thus far fallen short of identifying a reliable biomarker for ADHD. Despite some common trends, the genetic heterogeneity of ADHD and the fact that most individuals are not diagnosed until school age contributes to considerable variability in neural signatures among affected individuals. However, we believe that E/I imbalance, particularly in the frontal-striatal circuitry, may present a unifying, developmentally stable biomarker of ADHD. The challenge at this point will be to quantify E/I and test for short- and long-term associations with treatment response. We advocate for (1) the use of MR spectroscopy to measure effects of pharmacological treatment on excitatory and inhibitory neurotransmitters; and (2) characterization of changes in EEG 1/f spectral slope over the course of development, to serve as a normative comparison from which to quantify deviant trajectories associated with ADHD. These non-invasive, complementary approaches have both been used successfully in very young children as well as individuals with neurodevelopmental disorders. Most importantly, given the increasing prevalence of ADHD worldwide and its individual and societal impacts, these endeavors will be worth the financial and intellectual investment.

## Figures and Tables

**Figure 1 brainsci-11-00091-f001:**
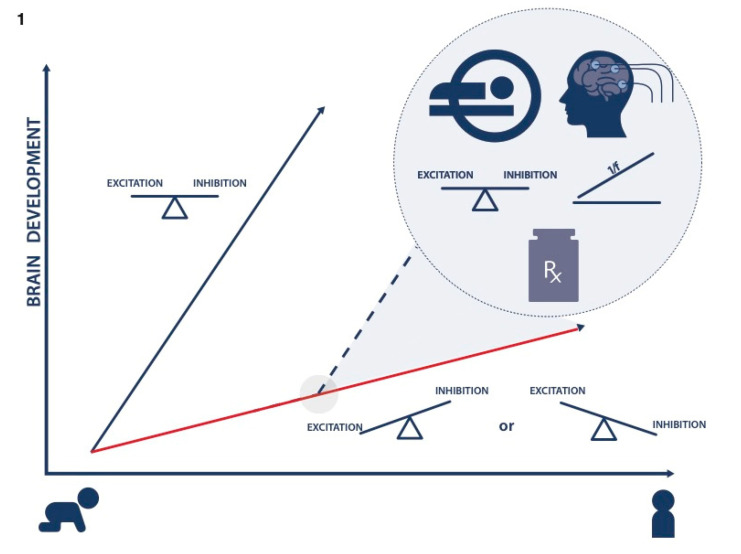
Precision medicine care for attention-deficit/hyperactivity disorder (ADHD). Red line represents atypical brain development and the blue line represents healthy brain growth. The dashed line represents projected healthy brain growth achieved by precision medicine care through early medical interventions guided by the measurements of excitation and inhibition (E/I) balance and 1/f ratio using MR spectroscopy (MRS) and electroencephalogram (EEG) methods.

## Data Availability

No data is available to share.

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
