# Peer review of "Precision Medicine Care in ADHD: The Case for Neural Excitation and Inhibition"

_brainsci, 2021, doi:10.3390/brainsci11010091_

Round 1
Reviewer 1 Report
First of all, thank you for the opportunity to review your manuscript.
Although the type of article submitted corresponds to the perspective, it would be convenient if the writing of the manuscript tries to answer different questions that make the authors' message clearer.
Likewise, it would be recommended that the authors make a graph, table or figure that summarizes part of the findings shown in their manuscript.
Given the importance of the topic and the amount of literature on the pathophysiology, neuroimaging and psychophysiology of ADHD, it would be recommended that the authors make an orderly synthesis of it.
On the other hand, it would be convenient for the writing to progress through neurodevelopment until adulthood, to see the importance of the circuits involved and the effects of the modulation of pharmacological treatment.
Finally, the part of the text that addresses the effect of glutamate and GABA modulation (lines 168 to 181) in other neurodevelopmental disorders, although it is sometimes comorbid with ADHD, does not show results of interest for the cognitive deficits shared with ADHD, for example, the restriction in social communication, repetitive behaviours, etc ..., which are more significant in ASDs. If these results are presented, they should be directly related to common neuropsychological deficits.
My recommendation for the future is that the authors carry out a systematic review and/or meta-analysis on the topic addressed.
Author Response
Dear reviewer #1:
Thank you very much for your helpful suggestions. We have attached our responses to your comments.
Kindest regards,
-Ping Chao Mamiya

Reviewer 2 Report
Comments to Authors:
The review manuscript “Precision Medicine Care in ADHD: The Case for 2 Neural Excitation and Inhibition” by Mamiya, Arnett. Stein is well-written and offers important insights into the physiopathology of ADHD, specifically by discussing neural circuits and release mechanisms associated with GABA, glutamate and catecholamines. Their conclusion is rather bold, as they propose, based on the literature review performed, that the excitatory and inhibitory systems and the catecholamine systems in the developing brain are both additive and interactive, supporting the notion that future therapies for ADHD could be developed based on modulation of glutamate/GABA balance.
The structure of the review is acceptable, but I have some suggestions to make the manuscript clearer to the readers:
- On Abstract, lines 16-18: the authors say they will discuss recent experimental evidence from “both animal and human studies”- instead, more appropriately they could say from both preclinical and clinical studies. Then, they go on to say that “the studies show an imbalance of E/I in the fronto-striatal circuitry in children and adults with ADHD”- meaning that they are focusing on the human studies, which exclude the previous statement on animal studies. To reconcile, I suggest splitting this sentence into 2 sentences.
- On Introduction, line 38, they mention “derivatives of methylphenidate and 37 amphetamine stimulant classes, as well as several non-stimulants”, but provide little to no explanation on the mechanism of these drugs. I suggest adding some information on their mechanism and appropriate interpretations.
- Still on the Introduction, line 55, the authors mention “very young patients with ADHD”, but do not provide an estimate on the age of these patients. Please provide this information.
- On lines 67-68, the authors state: “There is emerging evidence that GABA and glutamate levels and their ratio contribute to the 67 neurobiological etiology of neurodevelopmental disorders [18]”. Reference 18 is about Autism, so they need to be more specific.
- On line 68, they mention “several regulators, including GABA”- this could be misinterpreted, as GABA is a neurotransmitter itself, and part of the E/I balance, as they authors describe throughout the paper. So, what do they mean by GABA being regulator?
- On lines 71-72, they discuss computational studies. While the importance of such studies is underscored, one might wonder if the cited paper (25) discusses maturation of interneurons. Please clarify.
- On page 80, they talk about elevated Glx concentrations. Please define Glx.
- On line 96, the subtitle “Spectral slope” does not really convey the contents of the subsection, please revise. Still in this subsection, pages 111-113 provides information on “increased theta/beta ratio is a neural signature of ADHD, which would be consistent with a steeper 1/f slope [54].” This info should be provided in the beginning of the subsection, and then the authors can expand on their interpretation.
- On lines 149-150, the authors say: “The administration of stimulant medication is shown to normalize atypical brain connectivity measures among children with ADHD [70-73].” This present very broad information that need to be expanded and further discussed. They cite 4 papers to support this statement, there should be more information from these 4 papers that the readers could benefit if clearly presented here.
- Next, on line 150, they discuss altered GABA and glutamate concentrations, but they do not mention what the ‘normal’ concentrations would be, not even estimates. Please expand and discuss this further.
- On lines 168-170, they say: “GABA- and glutamate-modulators have already been tested for treatment of neurodevelopmental disorders that frequently co-occur with ADHD, such as epilepsy [80], autism spectrum disorder (ASD) [81], and Fragile X syndrome [82].” Which GABA and glutamate modulators are they referring to? How do they modulate? The readers would have to go to references 80-82 to find out. Please provide this information here.
- On line 172, they mention “GABA agonists”: what does this mean? Agonists of GABA receptors, or GABA transporters, or both, and which subtypes specifically? Please provide more information on the pharmacology here.
- On line 175, they mention bumetanide, on line 179, arbaclofen, on line 184, Gamalate B6- however, no information on the mechanism of these drugs is provided. Please expand on their pharmacology here.
- On line 186, the authors mention “metabotropic glutamate agonist, Org 26576 (8,9,9a,10-(S)-tetrahydro-5H, 7H-Pyrido-186 [3,2-f]pyrrolo [2,1-c][1,4] oxazepin-5-one)”- please be specific about the mechanism of this drug, which subtype of metabotropic receptor (include the word receptor) does this drug modulate? On lines 187-188, they say the drugs “was injected in adult rats significantly reduced locomotor activity in rats”- delete some of the ‘rats’, as it is redundant.
- On line 188, drug ‘Org” is mentioned without a definition of its mechanism of action.
- Line 190 mentions disruption of metabotropic glutamate receptor gene network- does this apply to all subtypes of mGluRs or some specific ones?
- Line 191 mentions a mGluR activator, which subtype of mGluR are the authors referring to?
- Line 192 mentions atomoxetine but not mechanism of action for this drug is provided.
- Line 201 mentions common stimulant medication for ADHD but does not provide which ones, be more specific please.
- The Conclusions and Future Directions section is nice and straight forward.
Author Response
Dear reviewer #2:
We thank you for your helpful suggestions. We have attached our responses to your comments.
Kindest regards,
-Ping Chao Mamiya

Round 2
Reviewer 1 Report
Despite the authors' modifications, the revised version does not have sufficient merit to be published. The figure added by the authors does not provide anything clarifying about the specific content of the manuscript.